# The Role of Nutritional Lifestyle and Physical Activity in Multiple Sclerosis Pathogenesis and Management: A Narrative Review

**DOI:** 10.3390/nu13113774

**Published:** 2021-10-25

**Authors:** Salvatore Fanara, Maria Aprile, Salvatore Iacono, Giuseppe Schirò, Alessia Bianchi, Filippo Brighina, Ligia Juliana Dominguez, Paolo Ragonese, Giuseppe Salemi

**Affiliations:** 1Department of Biomedicine, Neuroscience, and Advanced Diagnostics, University of Palermo, 90127 Palermo, Italy; salvatorefanara69@gmail.com (S.F.); dr.aprilemaria@gmail.com (M.A.); giuseppeschiro1994@gmail.com (G.S.); alessia.bianchi@unipa.it (A.B.); filippo.brighina@unipa.it (F.B.); ligia.dominguez@unipa.it (L.J.D.); paolo.ragonese@unipa.it (P.R.); giuseppe.salemi@unipa.it (G.S.); 2Neurology Unit, Department of Diagnostic and Therapeutic Radiology & Stroke, University of Palermo, 90127 Palermo, Italy; 3Neurophysiology Unit, Department of Diagnostic and Therapeutic Radiology & Stroke, University of Palermo, 90127 Palermo, Italy; 4Geriatric Unit, Department of Internal Medicine and Geriatrics, University of Palermo, 90127 Palermo, Italy

**Keywords:** multiple sclerosis, nutritional lifestyles, physical activity, low carbohydrate diet, Mediterranean diet, fasting-mimicking diet, Western diet, gluten-free diet, low-fat diet

## Abstract

Studies on the role of nutritional factors and physical activity (PA) in the pathogenesis of multiple sclerosis (MS) go back a long time. Despite the intrinsic difficulty of studying their positive or negative role in MS, the interest of researchers on these topics increased during the last few decades, since the role of diet has been investigated with the perspective of the association with disease-modifying drugs (DMD). The association of DMD, diets, and PA might have an additive effect in modifying disease severity. Among the various diets investigated (low-carbohydrate, gluten-free, Mediterranean, low-fat, fasting-mimicking, and Western diets) only low-carbohydrate, Mediterranean, and fast-mimicking diets have shown both in animal models and in humans a positive effect on MS course and in patient-reported outcomes (PROs). However, the Mediterranean diet is easier to be maintained compared to fast-mimicking and low-carbohydrate diets, which may lead to detrimental side effects requiring careful clinical monitoring. Conversely, the Western diet, which is characterized by a high intake of highly saturated fats and carbohydrates, may lead to the activation of pro-inflammatory immune pathways and is therefore not recommended. PA showed a positive effect both in animal models as well as on disease course and PROs in humans. Training with combined exercises is considered the more effective approach.

## 1. Introduction

Multiple sclerosis (MS) is a chronic, neuroinflammatory disease of the central nervous system (CNS) characterized by immune-mediated damage to oligodendrocytes with myelin and axonal damage [1]. A multigenic predisposing background interacts with environmental factors (i.e., microorganisms, passive smoking, toxic, ultraviolet radiation exposure), habits (active smoking), or individual phenotype characteristics (adolescent obesity), creating a dysregulation of the immune response with a higher pro-inflammatory response [2].

Often, the disease clinical onset or the further exacerbations are associated with other trigger factors such as infection, fever, anxiety, trauma, or puerperium.

In the last few decades, many treatments (i.e., DMD) have been identified, and their use improved the natural history of the disease, especially for the relapsing remitting form. However, DMD available at the present time are effective in specific subtypes and phases of MS as well as their use may be limited by lack of tolerance, side effects, or the presence of comorbidities with a large number of patients with unsatisfied needs.

The understanding of the role of nutritional factors and physical activity on the pathogenesis of MS and the opportunity to act by modulating these factors could be a complementary approach in disease management. The interest of researchers in these topics goes back a long time, but the intrinsic difficulty in studying their role, the limited resources allocated on this matter, as well as the tendency for the therapeutic use of incomplete data has made these issues confused. 

The aim of the present narrative review is to update existing evidence on nutritional lifestyle and physical activity from the perspective of MS management. We will also discuss the relationship between nutritional lifestyle (i.e., single diets, macronutrients, oligoelements) and MS pathogenesis and clinical course and the possible dietary approaches proposed to modify such a pro-inflammatory nutritional lifestyle. 

## 2. Nutritional Lifestyle as Risk Factor or Complementary Treatment for Patients with Multiple Sclerosis 

Despite a normal calorie intake, patients with MS (pwMS) often have an imbalance between macronutrients intake with low-carbohydrate and high-lipid diet associated with abdominal obesity, higher body mass index (BMI), waist-to-hip ratio, waist-to-height ratio, and higher fat percentage. This condition leads to a pro-inflammatory status with high serum levels of interleukin 6 (IL-6), TNF-alpha, and leptin, which are all related with MS pathogenesis [3,4]. Moreover, according to the existing evidence, there is a link between dietary intake of lipids and higher prevalence and MS progression [5]. 

Although MS is an autoimmune disease affecting CNS, the exact pathway triggering inflammation is still unknown. Recently, the commensal microbiota has attracted increasing interest regarding pathogenetic roles in immune-mediated diseases, and it acquired an important role in MS pathogenesis as a possible environmental risk factor. It seems to be involved in modulating the host’s immune system, triggering demyelination, altering the integrity and function of blood–brain barrier (BBB), and interacting with different cell types in CNS such as disease-inducing inflammatory T cells (e.g., Th1, Th17) [6]. In particular, the dysbiosis sustained by detrimental dietary habits (e.g., Western diet, low vitamin D intake, etc.) with increasing of phylum Firmicutes (e.g., Blautia, Dorea), some Bacterioides genera (Pedobacteria and Flavobacteria), Streptococcus oralis, Streptococcus mitis, Methanobrevibacter, Akkermansia, Proteobacteria, and decreasing of other Bacterioides genera (Prevotella, Bacterioides, Parabacterioides), Clostridium, Adlercreutzia, and Butyricinimonas might be associated to reduced anti-inflammatory cytokines production (e.g., IL-10). The results of these mechanisms would cause a reduction of anti-inflammatory metabolites such as propionate and butyrate, determining effects on Treg cells reduction, Th17 cells expansion (particularly at intestinal levels), upregulation of genes involved in antigen presentation to B and T cells, and activation of the complement and coagulation cascade with intestinal barrier dysfunction [7]. 

Therefore, the nutritional lifestyle leading to high BMI, abdominal obesity, or dysbiosis might induce a systemic pro-inflammatory state contributing to disease pathogenesis and severity. 

### 2.1. Food-Derived Factors

Several studies on the association between metabolic changes and the risk of developing MS or aggravating the disease course have been published. During the 1950s and 1960s, several studies were carried out considering the role of vitamin B12 in pwMS because of the suggested role of this vitamin in myelin formation. In 1996, Cari Loder proposed that a dietary supplementation with lofepramine, phenylalanine, and vitamin B12 could be effective in relieving the symptoms of pwMS [8]. In 1973, Mitchell and Schlandl gave an account of an association between pyridoxine deficiency and MS [9]. In 1974, Goldberg reported an association between low levels of vitamin D and calcium and the risk of developing MS, and a decade later, some author reported a reduction of disease activity through dietary supplementation with calcium, magnesium, and vitamin D [10,11]. A decrease in antioxidants and neuroprotective and immunoregulatory vitamins and an increase in total homocysteine (t-Hcy), cholesterol (CHL), and HDL cholesterol were reported in pwMS during the following decades [12]. 

The main food-derived molecules include the omega-3 polyunsaturated fatty acids (PUFAs), the polyphenols (e.g., epigallocatechin gallate, cocoa, etc.), oligoelements, and vitamins. 

It is known that dietary PUFAs such as eicosapentaenoic acid (EPA) and docosahexaenoic acid (DHA) have anti-inflammatory properties, whereas a high level of omega-6 fatty acids might promote the inflammatory process [13]. Both EPA and DHA improved EAE by reducing pro-inflammatory cytokines and pro-inflammatory cells (e.g., Th1, Th17, dendritic cells), whereas in RRMS, high doses of EPA and DHA reduced disability as assessed by the Expanded Disability Status Scale (EDSS) (Table 1). 

Moreover, the indication to introduce dietary anti-oxidant molecules such as polyphenols, carotenoids, and oligoelements (e.g., selenium, zinc, magnesium) in MS therapy derives from the observation that oxidative stress with the consequent generation of ROS plays a fundamental role in the pathogenesis of the disease [18] (Table 2 and Table 3).

The polyphenols group is divided into flavonoids (e.g., luteolin, baicalin, quercetin, cocoa) and nonflavonoids molecules such as resveratrol and caffeine.

Coffee consumption, although associated with an increased risk of developing rheumatoid arthritis and type 1 diabetes mellitus, has been supposed to reduce the risk of MS [53]. Indeed, coffee has negligible side effects and appears to be helpful for patients who experience fatigue related to the disease [40] (Table 2).

Finally, the relationship between hypovitaminosis D and the risk of MS has been largely explored. MS incidence and vitamin D deficiency are more common in countries with less sunlight exposure; low vitamin D levels at MS onset might be predictive of disease progression and higher long-term activity [54,55]. However, there are discordant data considering the daily supplementation of vitamin D in pwMS; some authors reported benefits of vitamin D supplementation in clinical (e.g., EDSS score, relapses) and radiological (i.e., brain MRI lesion load) outcomes, whereas a more recent study showed no effect of vitamin D on MS relapse rate or brain MRI activity [56,57]. Moreover, vitamin D intake above the upper limit of 4000 U/daily may mimic MS symptoms exacerbation, resulting in fatigue, urinary tract disorders, and weakness [58].

Although vitamin D is the only evidence-supported implementation among the vitamins in pwMS, it is important to avoid overall vitamin deficiencies. Indeed, B1, together with B6 and B12, showed neuroprotective effects after an episode of optic neuritis with improved visual parameters [59] (Table 4).

The complete description of the considered food-derived factors and vitamins is reported in Table 1, Table 2, Table 3 and Table 4.

### 2.2. Diets

#### 2.2.1. Low-Carbohydrate Diets

These diet habits consist in low daily carbohydrates intake such as ketogenic diet (KD) with its variants (e.g., modified Atkins diet, medium-chain triglyceride diet) and Paleolithic diet with its variants (e.g., Wahls elimination diet, Paleolithic modified diets).

The KD and modified Atkins diet are characterized by high fat intake with 4 g of fats per gram of carbs or proteins and less than 20 g of carbs daily intake respectively, resulting both in a fasting mimic state allowing metabolic shift from glycolytic energy toward oxidative phosphorylation by using fatty acids as the primary energetic substrates and elevated serum levels of ketones as consequence [80]. The ketone bodies (e.g., β-hydroxybutyrate) are transported across the BBB and might be used as energy and may have both anti-inflammatory and neuroprotective effects by upregulating the antioxidant pathway via Nrf2 and reducing pro-inflammatory cytokines production such as IL-1β [81,82]. Moreover, KD might allow recovering the colonic bio fermentative function, modulating the microbiome with an anti-inflammatory effect [83]. The benefits of KD were demonstrated not only in experimental autoimmune encephalomyelitis (EAE) with an improvement of memory and motor deficits but also in humans. Choi et al. described indeed the improvement in self-reported health-related quality of life and mild improvement in EDSS score in pwRRMS after six months of KD regimen [84]. A recent study suggested positive beneficial effects of modified Atkins diet in patients with RRMS showing improvement in BMI, EDSS score, non-dominant 9-hole peg test, and self-reported outcomes such as fatigue and depression as well as the reduction of serologic pro-inflammatory adipokines [85]. Although KD is safe, negative effects might include vitamin deficiencies, constipation, diarrhea, vomiting, weight loss, transient increases in blood lipides, liver steatosis, and acute pancreatitis [80].

The Paleolithic diet mimics the human ancestors’ diet before the agricultural and industrial revolutions involving the intake of non-domesticated meats, fish, vegetables, fruits, and legumes and avoidance of highly processed foods, dairy, and gluten, whereas the original modified Paleolithic diet (Wahls diets) recommends a high fruit and vegetables intake (1950 g+) divided among leafy, sulfur-rich, and deeply colored, encouragement of seaweed, algae, and nutritional yeast consumption; allowance for limited servings of gluten-free grains (e.g., rice) and legumes (e.g., soy milk); elimination of eggs, which are conversely allowed on a Paleo diet, and lower meat/fish intake [86]. Moreover, in 2015, the Wahls elimination diet also eliminated lectin-rich foods such as gluten-free grains, legumes, and nightshades (e.g., tomatoes, white potatoes, eggplant, peppers, and seed spices), as they may contribute to intestinal permeability and central nervous system inflammation [86]. Studies investigating the effects of the Paleolithic diet in pwMS are limited to the Wahls diet that showed improvement in quality of life, fatigue, gait, balance, anxiety, depression, and executive functions in patients with progressive MS [87]. More recently, an interventional randomized trial among modified Paleolithic diet versus usual care in patients with relapsing–remitting MS (RRMS) reported higher serum levels of vitamin K, which might reduce inflammation; improvements in fatigue severity scale and MS quality of life as well as in the non-dominant 9-hole-peg test and the 25-foot walk test time were found [88]. However, the avoidance of cereals in modified Paleolithic diets might lead to folic acid, thiamine, calcium, and vitamins B, D, and E deficiencies [89].

#### 2.2.2. Gluten-Free Diet (GFD)

Gluten is a one of the major components of the Western diets, and it is able to induce an immune-mediated small intestinal enteropathy know as celiac disease (CD), although there is increasing evidence that gluten might affect the human body by increasing gut permeability, activating the innate immune system, and increasing BBB permeability, making a cross-reaction with neural protein and activating autoreactive T cells [55]. However, some authors reported a higher prevalence of anti-gliadin immunoglobulins (AGA) or immunoglobulin against tissue transglutaminase (IgA-tTG, IgG-tTG) in pwMS, whereas similar antibody levels between pwMS and healthy controls were revealed by others [90]. Furthermore, several studies investigated the prevalence and the risk to develop MS in patients with CD failing to show a significant association [91]. The benefit of GFD with respect to a free diet among pwMS has been investigated only by a single, non-randomized trial, suggesting a significant lower activity on magnetic resonance imaging (MRI) and lower EDSS scores [92].

The “Wahls Protocol”, finally consisting of a multimodal lifestyle intervention, includes elimination of gluten from the diet with improvement of some self-reported outcome such as mood, fatigue, and quality of life in pwMS [93].

As a result of the limitation of the clinical trials mentioned above and the multimodality nature of Wahls’ protocol that does not allow assessing the exact benefit of a gluten elimination diet together with the controversial role of gluten in MS pathogenesis, up to date, no evidence is strong enough to recommend GFD in pwMS.

#### 2.2.3. Mediterranean Diet (MD)

The Mediterranean diet (MD) consists of favoring omega-3 polyunsaturated fats from fish or seafood (≥2 servings per week), monounsaturated fats from olive oil (at each meal), consumption of fruit (1–2 servings per meal), vegetables (≥2 servings per meal), bread, and cereals (1–2 servings per meal), moderate amounts of red wine, eggs (2–4 serves weekly), legumes (≥2 servings per week), and nuts (1–2 servings per day) [94]. The protective effects of MD on chronic diseases (e.g., cardiovascular, oncological, hypertension) and on waist circumference as well as on glucose and lipid metabolism are well known [95]. In MS, the MD is supposed to act directly through modulation of the chronic inflammatory state or indirectly reducing chronic comorbidities [96].

In a case-control study involving 70 pwMS, adherence to a MD reduced MS risk, and specifically, the consumption of high amounts of fruit and vegetables and the higher refined grains consumption were associated inversely and positively, respectively with increased risk of MS [97].

The number of relapses does not seem to be influenced by the Mediterranean diet, while the age of onset of the disease shows a weak correlation with a more delayed time of onset of the disease in patients who consume foods that are part of the Mediterranean diet [98]. However, giving the role of MD in chronic inflammation, it is reasonable that MD affects the long-term autoimmunity and secondary neurodegeneration with a beneficial effect on MS course and on long-term disability rather than the effect of reducing relapses during the active inflammation phase [98]. Furthermore, the beneficial role of MD on long-term disability may also be mediated by the positive effect on gut microbiota, thus reducing the persistent intestinal inflammation (see Western diets) and limiting the spreading of chronic systemic inflammation to the CNS [98].

Moreover, the beneficial effects of the MD appear to derive in part from the availability of polyphenols in foods, as suggested by the fact that in a rat model, dietary supplementation with olive leaf extract, which possesses large amounts of polyphenols, led to an observed reduced severity of EAE [99].

In light of these results, the Mediterranean diet is considered a safe type of diet and can be considered as a therapeutic measure to improve the quality of life of pwMS.

#### 2.2.4. Low-Fat Diet: Swank and McDougall Diets

As Dr. Swank in 1950 first noted that high saturated fat-enriched diets might increase the risk of developing MS, he proposed a low saturated fat diet excluding processed foods, high-fat diary, and red meat [100,101]. The Swank diet includes also daily intake of unsaturated fat, 5 g of cod liver oil, 10–15 g of vegetable or fish oil, less than 3 eggs weekly, no more than 3 cups containing caffeine per day, whole wheat bread, and fish more than 3 times weekly. In addition, as the person strictly adherent to the Swank diet may be at risk for vitamin A, C, E, and folate deficiencies, there is a recommended supplementation of B12 (1000 µg), folate (1000 µg), and vitamin D (5000 IU) [102].

Swank monitored the effects of the dietetic regimen over a 50-year period reporting fewer relapses in pwMS that strictly followed the diet (≤20 g daily saturated fat) and a three times lower risk of dying in comparison with pwMS who did not follow the diet [103].

However, the Swank diet has not been officially accepted due to the lack of the control group among studies; however, an RCT between the Swank diet and modified Paleolithic diet (Wahls elimination diet) is ongoing.

The McDougall diet is another low-fat diet (very low-fat) that is rich in fruit and carbohydrate, plant-based (vegan), low in sodium, without animal products (e.g., diary, eggs), and with oil addition, leading to a reduction of cardiovascular and metabolic diseases risks [104]. The McDougall diet in pwRRMS was associated with less serum cholesterol, insulin, BMI, and fatigue severity scale without any changing in EDSS score and brain MRI outcomes compared to healthy subjects [105]. However, persons strictly following the McDougall diet may present nutritional deficits such as iron, zinc, vitamin B12, vitamin D, calcium, and omega-3 [106].

#### 2.2.5. Fasting-Mimicking Diets (FMD)

FMD does not refer to a specific diet protocol, but it includes a myriad of diet interventions leading to a fasting state through a prolonged period of little or no caloric intake. In particular, common fasting protocols include severe calorie restriction (CR) and intermittent fasting diet (IF), whereas mild daily calorie restriction does not mimic a fasting state [107].

Since it is difficult to deal with a severe CR diet, especially for long periods, the IF diet has been proposed as a valid alternative including alternate-day fasting, the 5:2 diet with twice-weekly fasting, intermittent calorie restriction (ICR, severe calorie restriction two times weekly), or time-restricted feeding (limiting the whole daily caloric intake within an ≤8 h window) [108].

The calorie restriction (CR) diet, at 33% of daily calories, showed an anti-inflammatory effect by inhibiting pro-inflammatory cytokines (e.g., TNF-α, IL-6) [109] as well as a neuroprotective effect such as the reduction of demyelination in mice by enhancing the expression of brain-derived neuron factors (BDNF), Sox-2, and Sirt-1, which overall allow oligodendrocytes differentiation [110].

An in vivo model on EAE showed that severe CR to 66% is able to suppress neurological signs related to inflammation by increasing ACTH and corticosterone plasma levels with impaired interferon-γ production [111].

Moreover, FMD with cycles of three days of CR followed by four days of normal calorie intake was associated with EAE severity improvement with increased corticosterone and regulatory T cells, reduction of myelin loss, and lower inflammatory cells infiltrates in the spinal cord [112].

Furthermore, IF is able to improve EAE by inducing protective changes in the gut microbiota with an abundance of Bacteroidaceae, Lactobacillaceae, and Prevotellaceae with reduced leptin and B-cells levels in pwMS after 15 days compared to free-diet controls [113].

Both CR and intermittent CR in pwMS showed improvement in emotional health, suggesting a role for diet in depression with greater adherence of patients who followed intermittent CR [114].

Finally, one FMD cycle followed by MD showed improvement of self-reported quality of life and EDSS score in 20 patients with RRMS compared to controls and patients following KD, although both CR and an adapted ketogenic diet were associated with a reduction of pro-inflammatory enzymes in pwMS [84,115].

A recent study comparing the safety and feasibility of IF, continuous calorie restriction, and TRM in 90 pwRRMS reported poor adherence to the three diets without significant differences in fatigue, quality of life, and sleep quality among patients [107]. Although FMD showed promising results in EAE and in limited studies among pwMS, these diets are very hard to adhere to, limiting the effectiveness of the studies, and they might lead to side effects such as reduction of both bone mass and libido or menstrual cycles abnormality [116].

#### 2.2.6. Western Diets (WD)

Western diets are characterized by a high saturated fats and sugar-rich daily intake, which are commonly found in processed food, with a high fat/carbs or protein daily ratio leading to low-grade chronic inflammation, obesity, cardiovascular disease, hyperlipidemia, and type 2 diabetes, which could overall negatively impact in health and affect the disease course [117]. WD might also lead to dysbiosis in the gut microbiome with enteric inflammation, epithelium permeabilization, and endotoxemia worsening CNS inflammation [96]. Indeed, long-chain fatty acid exacerbated EAE severity by expanding Th1 and Th17 cells in the small intestine [118]. WD includes high-salt intake, whose effects on MS are controversial (Table 3). Moreover, some authors reported that a high-fat diet consumption was associated with EAE exacerbation and exaggerated autoreactive immune responses and neuroinflammation [119].

In pediatric MS, high saturated fat diets were associated to higher MS disease activity with each 10% increase in saturated fat being associated to a three-fold increased relapse risk. It was reported that pwMS following a healthy diet regimen were associated with a 25% reduced risk of clinically isolated syndrome compared to those adhering to a Western diet [120,121].

Finally, a higher proportion of fibers as the high nutrient quality carbohydrates (e.g., whole grain products) and a lower proportion of fat are able to improve physical activity and fatigue in pwMS as recently reported, whereas a high intake of sugar is related to an elevated serum insulin level, leading to inflammation and MS severity worsening [122].

A summary of the changes on inflammation pathways by the food and food-derived molecules being part of the above-described diet habits is reported in Figure 1.

## 3. Clinical Impact of Exercise in Patients with Multiple Sclerosis

Several studies investigated the effects of physical activity on MS through assessment of motor ability (e.g., mobility, walking), white matter integrity, cognitive functions, fatigue and mood, with controversial results.

Up to 70% of pwMS experience cognitive impairment, involving mostly memory and processing speed [123], as well as verbal fluency, visual learning, and attention [124]. Since few DMD showed a benefit effect on cognitive functions in pwMS [125], recently, the researchers started to focus on exercise training as a non-invasive treatment option for MS, taking into account the neuroprotective effect of physical activity in healthy and cognitively impaired older subjects.

A growing body of literature has begun to investigate physical exercise effects, differentiating high-intensity interval training (HIIT) from moderate continuous exercise (MCE) [126]. However, to date, different trainings such as yoga, Pilates, resistance, or aerobic exercise showed conflicting results.

The cognitive performance in pwMS is internationally assessed through the Brief International Cognitive Assessment for MS (BICAMS), within which the Symbol Digit Modalities Test (SDMT) was correlated with cognitive impairment, brain lesion burden, and brain atrophy [127].

In a recent study, Mayo et al. found a significantly improvement of SDMT scores in pwRRMS after 12 weeks of speed walking [128]. The authors investigated also structural brain changes other than clinical response (i.e., cognition, fatigue, and mood). Supposing a neuroprotective effect of physical exercise, these authors used diffusion tensor imaging (DTI) to detect microstructural changes of white matter. They did not find any white matter changes, neither improvements nor declines in the integrity, after 12 weeks of speed walking [127].

Improvement of processing speed was reported, as well, in a study based on analysis of cardiorespiratory fitness in two groups (HIIT vs. MCE), observing significantly improved effects in both groups [126].

However, not all published studies used the BICAMS tests. In a randomized controlled trial, participants underwent a supervised long-term high-intensity progressive aerobic exercise (PAE) performed during a 24-week period, and after, they were assessed through the Brief Repeatable Battery of Neuropsychological tests (BRB-N). When compared with a waitlist group, the cognitively impaired subgroup showed an improvement of the SDMT and Selective Reminding Test (SRT) after PAE. No effects were found in other tests both in the cognitively impaired subgroup and in the whole group, failing to show a clear correlation between changes in cardiorespiratory fitness and global cognitive performance in pwMS [129].

However, studies investigating physical exercise effects on cognitive functions showed conflicting results. No improvements in memory and learning were found after 26 weeks of supervised yoga/aerobic exercise [130] and after 12 weeks of moderate aerobic exercise [131]; in the same way, no improvements in processing speed were found after 8 weeks of Pilates [132].

Most of the researchers highlighted critical issues explaining these conflicting results such as methodological limitations such as inadequate samples, the fact that the enrolled patients did not always have an overt cognitive impairment [132,133], unsupervised sessions of several exercise modalities, low intensity and short duration of exercise (<12 weeks) [132], along with the application of single or several cognitive tests rather than a unique and validated neuropsychological battery [129]. To date, direct comparison between studies is challenging and, when compared, divergent results are obtained.

However, based on the most recent knowledge, there is general agreement that a combined exercise training in MS patients to achieve different effects is recommended [134].

A study by Ozkul et al. documented an increase in BDNF in serum after 8 weeks of combined exercise, pointing out the promoting cognitive functions of training [135].

A study performed in 2020 showed the beneficial effects of 8 weeks of Pilates and aerobic combined training in mild disabled pwRRMS with cognitive impairment, especially in cognitive functions, mood, mutual relationship, and quality of life. The “Exercise group” showed significant group-by-time interactions on long-term verbal memory, cognitive fatigue, quality of life, and walking capacity (*p* < 0.003). Processing speed, visuospatial memory, and verbal fluency were also improved in the group treated with combined exercises (*p* < 0.05). Conversely, the “control group”, exposed to relaxation exercises at home alone, showed only an increase in verbal memory scores [136].

### 3.1. Biological and Radiological Modifications Induced by Physical Activity in EAE/MS

#### 3.1.1. Pre-Clinical Studies

These studies showed how moderate/vigorous physical activity, practiced before and/or after the induction of the disease, positively influences the pathogenetic mechanisms involved in demyelination and neurodegeneration, which are hallmarks of the MS pathogenesis.

BDNF is a neurotrophin produced by microglia promoting neuronal survival and synaptogenesis through interaction with tyrosine kinase receptor B (TrkB) [137]. Some studies have shown a BDNF increase after exercise in animal models of EAE. Bernardes et al. observed increased BDNF levels in both brain and spinal cord in EAE mice after exercise [138]. Xie et al. showed an increase in BDNF in the CNS after high-intensity swimming training before and after the induction of the disease [139]. Interestingly, this is a common pathway, with one of the DMTs approved for MS (i.e., glatiramer acetate). The mechanisms underlying exercise-induced BDNF level elevation are still unclear. It has been hypothesized that neuronal hyperactivity and elevation of brain–blood flow might play a central role in BDNF increasing, as highlighted by enhanced level of BDNF, c-fos (marker of neuronal activity), and peNOS (i.e., endothelial nitric oxide synthase phosphorylated; marker of elevation in brain blood flow) during physical activity in a intensity-dependent manner [140]. The increase in BDNF-specific mRNA induced by long-term exercise, the restoration of glucose tolerance and the phosphorylation state recovery of TrkB receptors have also been described as potential mechanisms of exercise-induced BDNF elevation [141].

Moreover, several studies showed the effect of exercise in changing cytokines and cell populations of adaptive and innate immunity at the CNS level. In particular, exercise is able to reduce the levels of pro-inflammatory cytokines (IFN-γ, IL-17, and IL-1β), increase the concentrations of IL-10 and TGF-β (anti-inflammatory cytokines), as well as determine a reduction of Th1 and Th17 cells and an increase in Treg cells [139,142].

Exercise is able to determine a lower activation of microglia in cuprizone-induced demyelination [143].

Changes in the immune system at the level of peripheral lymphoid tissue have also been highlighted. In the splenic animal tissue, a significant increase in Treg cells was observed after exercise, together with an increase in IL-10 levels and reduction in the concentrations of IL-6, MCP-1, and TNF-α (cytokines of innate immunity) [142]. Moreover, the upregulation of Treg and downregulation of antigen-specific T cell proliferation and Th1 and Th17 populations from draining lymph node cells was highlighted [139].

The above-mentioned changes in the innate and adaptive immunity might be related to a wide production of glucocorticoids, catecholamine, and prostaglandin E2 in response to physical activity. For instance, glucocorticoids are able to reduce IL-12, IFN-γ, and IL-17, whereas PGE2 might suppress T cells realizing IFN-γ and IL-17 [139]. Furthermore, it has been shown as physical activity reduces the ability of dopamine to enhance Th17 cytokines or suppress IL-10, whereas serotonin was almost anti-inflammatory by reducing IFN-γ, IL-6, and TNF-α [144].

Together with an anti-inflammatory effect, physical activity also showed antioxidant properties. After exercise, in EAE, Souza et al. observed a reduction of lipid peroxidation, protein oxidation, and NO levels together with an increase in glutathione levels, glutathione peroxidase activity, and Nrf2, which is an important transcription factor involved in the antioxidant response (Nrf2 is the target dimethyl fumarate) [142].

The exercise also was able to counteract BBB changes associated with MS and EAE. In particular, a greater expression of tight-junction proteins and a decreased expression of adhesion molecules have been demonstrated in MS, leading to a reduced transmigration of inflammatory cells through the BBB [142]. Furthermore, in mice after exercise, an increased number of pericytes in blood vessels (which can promote the proliferation of oligodendrocyte progenitors in demyelinating lesions through the secretion of laminin α2) and an increased angiogenesis have been shown; both effects may promote remyelination [145].

Physical activity in EAE has been shown to restore glutamate levels at the pre-synaptic cortical level, which are markedly reduced after the induction of the disease and associated with the onset of symptoms [146].

Voluntary or forced exercise, before or after induction, is able to reduce the brain genetic pathway inhibiting axonal regeneration (Nogo-A, NgR, ROCK), being the target of the first agent introduced to restore cell damage in MS patients (opicinumab, an anti-LINGO-1 antibody) [147].

In addition, physical activity in the animal model promotes the mobilization of neural progenitors toward demyelinating lesions, where they tend to differentiate into oligodendrocytes [148]. Jenesen et al. demonstrated that exercise promotes oligodendrogenesis via the broad activation of pro-remyelination pathways, including the clearance of inhibitory lipid debris from lesions, emphasizing the role of activation of the peroxisome proliferator-activated receptor gamma co-activator 1-alpha (PGC1α) within oligodendrocytes; remyelination by exercise was bolstered when combined with the regenerative medication clemastine [149]. The beneficial effects of the physical activity on CNS are believed to be both central (mechanisms that primarily involve the brain such as increasing neuronal activity) and peripheral (e.g., myokines released by skeletal muscles may modulate neurotrophic factors levels in CNS) [145,150].

The above-mentioned mechanisms may at least explain the effects of physical activity on the pathological hallmarks of the MS. Indeed, significant effects on demyelination, immune cell infiltration, and neurodegeneration have been demonstrated [138,139,143,147,150,151].

However, the correlation between clinical improvement and decrease in CNS inflammatory infiltration was not demonstrated [138], and no benefit in reducing demyelination was found [152].

The histopathological changes into CNS were correlated with benefits on the course of the disease such as delay disease onset, better clinical scores, and minor body weight loss [138,139,140,141,142,143,144,145,146,147,150,151].

To our knowledge, only one study reported no clinical benefit [153].

In these studies, exercise is generally done before or shortly after the experimental induction of the EAE, and the benefits were found regardless the type of exercise (e.g., aerobic, resistance training, increased activity). However, one study suggested that high-intensity exercise may have greater benefits than the same exercise at moderate intensity on clinical and pathological outcomes [139].

Such evidence derived from basic research leads us to believe that moderate-vigorous physical activity can be an effective and safe non-drug treatment to be implemented early and concurrently with DMD in MS patients. In this regard, Lozinski et al. introduced the concept of “MedXercise”, referring to the fact that eXercise may create a favorable microenvironment in the CNS for concomitant regenerative Medication to increase its reparative potential, maximizing the recovery of lesions: it is the case of the above-mentioned synergy effect between exercise and clemastine on remyelination [145].

#### 3.1.2. Clinical Studies

Clinical studies performed in pwMS reveal minor or negligible effects on systemic levels of cytokines and neurotrophic factors [150].

This discrepancy can be mainly explained by the fact that the evidence derived from preclinical studies is based on assessments at the tissue level, so it is not surprising that such mechanisms are not in place at the systemic level. Furthermore, studies with pwMS are often of short duration (may not induce chronic effects) and include a small number of individuals [150]. For example, in a recent study, exercise decreased plasma neurofilament light chain concentration and rerouted the kynurenine pathway only in short-term evaluation and not after 3-week training [154].

Despite these limitations, recent studies showed an improvement in MS-related cytokine levels and neurotrophic factors after a 12-week exercise program, regardless of the disability [155,156].

Some ongoing trials may gave us more information from this perspective, such as the Exercise PRO-MS study. This was a 48-week phase II study on patients with progressive MS in which there was a planned evaluation of BDNF and neurofilament blood levels as secondary outcome [157]; an RCT is also underway that has as its primary outcome the assessment of serum concentrations of MMP-2 after 12 weeks of exercise [158].

The main evidence of the effect of physical exercise on the course of the disease in pwMS derived from imaging studies. Indeed, several studies showed an association between brain plasticity [159], specific brain structure volumes, and structural and functional connectivity [145,157,160,161,162]. These results are important, as they are obtained after a short period of time, so further long-term studies are needed to understand what the effect of exercise may be in delaying neurodegeneration [150].

However, the studies performed to date have several limitations (e.g., short duration, trials not specifically designed to evaluate PA outcomes) with weak clinical evidence as a consequence.

Few studies have been performed considering the role of physical activity as a primary prevention factor in reducing the risk of MS, and basing on these, Dalgas et al. proposed the “exercise-induced postponement theory”, according to which regular moderate to high intensity exercise can postpone the clinical diagnosis of the disease [150]. However, studies suggesting that exercise could postpone the occurrence of disease activity and progression in pwMS have severe limitations, and further studies are needed to prove it [150,161,163]. It should be underlined that almost all clinical studies, focusing mainly on the symptomatic effect, have included patients with a long disease duration; thereby, the effect of exercise in this “window of opportunity” has not yet been studied, as it is conversely occurs in animals [150].

The Early Multiple Sclerosis Exercise Study (EMSES) is an ongoing RCT with the goal to overcome the above-mentioned limitations. In EMSES, the 48-week exercise therapy will be evaluated as a supplemental treatment in early MS, primarily investigating the disease activity (e.g., relapse rate) and neurodegeneration (e.g., brain atrophy) and secondarily the progression of symptoms and disability [164].

### 3.2. Recommendations

While in the past, exercise was a controversial treatment as it was thought to exacerbate the symptoms of MS, today, PA is considered not only safe without serious adverse effects (as opposed to conventional DMDs) but also beneficial on disease symptoms and potentially capable of acting as a disease-modifying therapy (Figure 1). Therefore, PA should be prescribed early along with medical treatment in pwMS [150], while more often, it is prescribed as a symptomatic treatment after many years of illness [164]. Based on current evidence and expert opinion, the National MS Society recently made recommendations for clinicians who treat pwMS on promoting exercise and lifestyle physical activity across disability levels [165]. Healthcare providers should encourage ≥150 min/week of physical activity, and early evaluation by a specialist is recommended in order to ensure individualized treatment.

## Figures and Tables

**Figure 1 nutrients-13-03774-f001:**
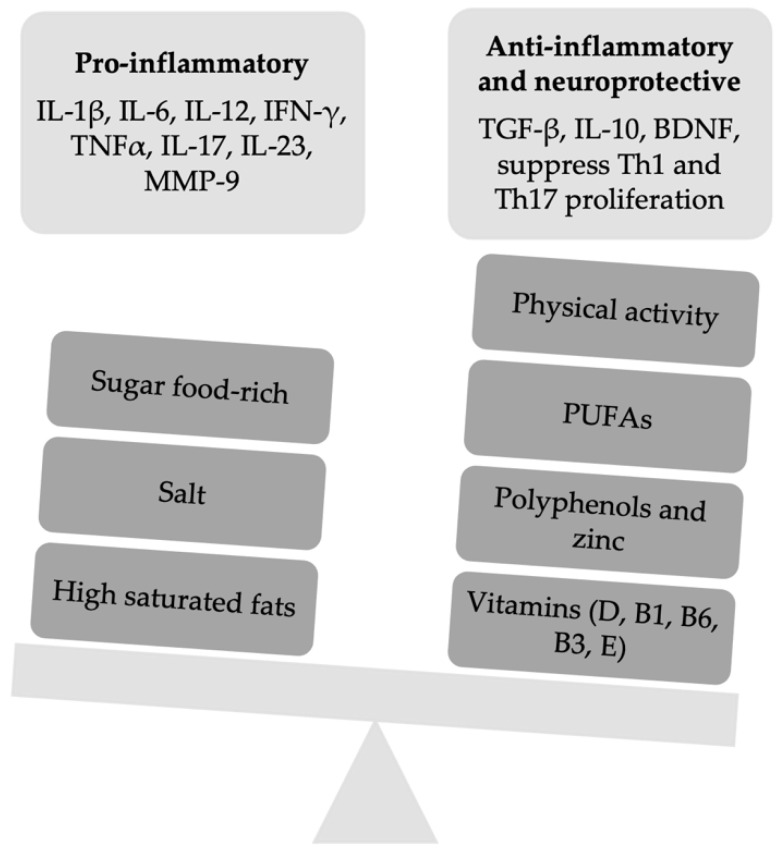
A summary of the slight imbalance between pro-inflammatory and anti-inflammatory factors of the innate and adaptive immune response. IFN, interferon; Th, T helper cells; MMP-9, metalloproteinasis-9; TGF, transforming growth factor; BDN, brain-derived growth factor; PUFAs, omega-3 polyunsaturated fatty acids.

**Table 1 nutrients-13-03774-t001:** Molecular mechanism and benefit on pwMS of the food-derived PUFA.

Food-Derived Factor	Molecular Mechanism Proposed	Benefits on PwMS	References
PUFAs	EPA improves EAE by reducing IFN-γ and IL-17 production, specifically in CNS lesions infiltrated by CD4 T cells, and it enhances PPARs. DHA reduces the expression of transcription factors in DCs for Th1 and Th17 differentiation as well as the mortality in mice with EAE	Omega-3 improve the metabolic profile, EDSS score, and reduces inflammatory markers such as MMP-9, TNF α, IL-1β, and IL-6	[14,15,16,17]

PUFAs, omega-3 polyunsaturated fatty acids; EPA, eicosapentaenoic acid; DHA, docosahexaenoic acid; IL, interleukin; TNF, tumor necrosis factor; MMP-9, metalloproteinasis-9; DCs, dendritic cells; EAE, experimental autoimmune encephalomyelitis; CNS, central nervous system; EDSS, expanded disability status scale.

**Table 2 nutrients-13-03774-t002:** Molecular mechanism and possible advantages on pwMS of food-derived polyphenols.

Food-Derived Factor	Molecular Mechanism Proposed	Benefits on PwMS	References
Luteolin	Suppress the migration of PMBCs in animal model and prevent disease relapses by influencing the monocytic GTPase activity.	Reduction of EAE severity with protective effect on chronic EAE. Dose-dependent anti-inflammatory effect by reducing IL-1β, TNF-α, and MMP-9. The therapy with IFN-β has a summation effect with that of luteolin.	[19,20]
Baicalin	Reduces oxidative stress in myelin-producing cells through the Nrf2/HO-1 signaling pathway Reduces CNS inflammation by suppressing IL-17, IFN-γ, GM-CSF, IL1-β, IL-6, IL-1, and IL-23	Improvement in the clinical score. Stopping baicalin results in recurrence of symptoms 7–8 days after the treatment.	[21,22]
Curcumin	Improves EAE through inhibition of the STAT3-phosphorylation and reduction of IL-12 production from microglial cells In PMBCs, suppressed IFN-γ and increased IFN-β Reduced production of MMP-9 in human astrocytes.	A dose of 80 mg daily for six months in pwRRMS reduced pro-inflammatory pathways compared to control without improving EDSS score. It might potentially reduce BBB permeability ameliorating MS clinical manifestation.	[23,24,25,26,27]
Resveratrol	In EAE, decreases the production of pro-inflammatory cytokines (e.g., TNF-α, IFN-γ, IL-2, IL-9, IL-12, and IL-17), induces T regulatory cells, and decreases in a dose-depended manner the BBB disruption by reducing the loss of TJ components (e.g., claudin-5, occludin). Contrasts EAE development through suppression of the miRNA-124/SK1 pathway. Neuroprotection during optic neuritis in an EAE model by reducing axonal loss.	Although 150 mg daily resveratrol supplementation in association with vitamin D in pwMS showed a reduction in serum levels of MMP-9, it did not demonstrate an improvement in signs and symptoms	[28,29,30,31,32,33]
Epigallocatechin gallate	Modulates GABAergic pathway. Improves EAE severity by reducing Th1 and Th17 cells.	It reduces IL-6, improving anxiety and depression. More in men than women, it improves the energy metabolism during exercise without any clinical and radiological effects	[34,35,36,37,38]
Cocoa	Increases cerebral blood flow. Antioxidant properties alleviate lipid peroxidation and axon damage	Mild reduction in fatigue and fatigability	[39]
Caffeine	In EAE after immunization with 10–30 mg/kg daily, caffeine reduces inflammatory cells in the spinal cord and neurological as well as IFN-γ production and disease severity.	An estimated amount of 250 to 300 mg caffeine intake (2–3 cups) improves fatigue and mental capacity, especially in patients with an EDSS higher than 0 but lower than 4	[40,41,42]
Alcohol	In EAE, male-specific disease remission induced potentially via gut microbiota modulation.	Not available	[43]

IL, interleukin; TNF, tumor necrosis factor; MMP-9, metalloproteinasis-9; EAE, experimental autoimmune encephalomyelitis; CNS, central nervous system; EDSS, expanded disability status scale; PMBCs, peripheral blood mononuclear cells; Nrf2, NF E2 related-factor 2; HO-1, heme-oxygenase-1; IFN, interferon; GM-CSF, granulocyte-macrophage-colony-stimulating factor; TJ, tight junctions; SK-1, sphingosine kinase 1.

**Table 3 nutrients-13-03774-t003:** Molecular mechanism and possible advantages on pwMS of food-derived oligoelements.

Food-Derived Factor	Molecular Mechanism Proposed	Benefits on PwMS	References
Zinc	Medium dose (1.5 mg/kg) improves clinical score of EAE, suppressing T cell activation and pro-inflammatory cytokines, whereas high doses (6 mg/kg) lead to clinical worsening.	Zinc supplementation (220 mg zinc sulfate daily) is able to improve depression in MS patients without providing any benefit for movement disorders.	[44,45]
Salt	High-sodium intake might inhibit the functions of Treg cells, promoting the shift to a Th1-like phenotype. High-salt intake induces EAE exacerbation through changes in microbiota and enhanced Th17 cells differentiation. High-salt intake might enhance corticosterone serum levels, allowing the expression of TJ molecules suppressing CNS autoimmunity.	Some authors reported that salt intake did not have a relationship with the course or brain MRI activity of MS, whereas other ones reported that high-sodium intake is associated with increased MS relapses and brain MRI activity. However, in pediatric onset MS, no associations between salt intake and pediatric onset MS risk or time to relapses were found.	[46,47,48,49,50,51,52]

Th, T helper cells; EAE, experimental autoimmune encephalomyelitis; TJ, tight junctions.

**Table 4 nutrients-13-03774-t004:** Molecular mechanism and possible advantages on pwMS of food-derived vitamins.

Food-Derived Factor	Molecular Mechanism Proposed	Benefits on PwMS	References
Vitamin D	In the EAE model, vitamin D3 suppresses Th17 and Th1 differentiation, enhancing the percentage of Treg cells. Reduces demyelination, incidence, and clinical score of EAE Increases the effectiveness of steroid therapy through mTORC1 inhibition.In EAE, high doses of vitamin D leading to hypercalcemia might promote T cells proliferation with clinical exacerbation	Vitamin D3 supplementation does not improve the depressive symptoms in MS. It enhances the response to some treatments for MS such as interferon-β	[60,61,62,63,64,65,66,67,68,69]
Vitamin B1	In EAE, vitamin B1 deficiency causes Th1 and Th17 spinal cord infiltration	PwMS with reduced vitamin B1 levels and depression might benefit from supplementation (potential role in fatigue severity improvement)	[70,71,72]
Vitamin B3	Potentiates beneficial effects on monocytes and macrophages in promoting remyelination in CNS aging	No data available	[73]
Vitamin B7	It promotes myelin synthesis, and it reduces axon hypoxia in MS	Some authors reported benefits in worsening EDSS score in patients with progressive MS High doses might increase the risk of relapses	[74,75,76]
Vitamin A	Antioxidant	It might improve fatigue and depression during interferon therapy	[77]
Vitamin E	Hippocampal remyelination, and it suppresses INF-γ production and delays EAE progression	Not available	[78,79]

CNS, central nervous system; EDSS, expanded disability status scale; IFN, interferon; Th, T helper cells; EAE, experimental autoimmune encephalomyelitis.

## Data Availability

Not applicable.

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
