# Peer review of "The Role of Nutritional Lifestyle and Physical Activity in Multiple Sclerosis Pathogenesis and Management: A Narrative Review"

_nutrients, 2021, doi:10.3390/nu13113774_

Round 1

Reviewer 1 Report

The article is quiet interesting and brings under the light an interesting alternative to drugs.

 I would recommande to discuss further about innate immunity and adaptive immunity. What are the pathways involved in changes that studies are reporting ? Why inflammatory pathways are down regulated, 

BDNF role is still debate as mentioned in this brief review PMID:33173466. however, it seems to have beneficial role on brain recovery. In the review you talk about BDNF and exercise but what could stimulate the production of BDNF during exercice.

Besides, some sentences are not clear

108 " properties whereas an high omega- 6 fatty acid might promote inflammatory process ..." high level of mega??? 

258 "Swank monitored the effects of its dietetic regimen among 50 years reporting fewer  relapses in pwMS that strictly followed its diet" what do you mean by 50 years? I guess it's the age of subjects... 

330  "...25% rieduced risk..."

Mediterranean diet section is quiet interesting but it should be developed. 

It would be helpful to add a figure that sum-up everything. 

Author Response

We thank for your advice useful for improving our manuscript.

As suggested, we took the opportunity to better explain how physical activity affects innate and adaptive immune response. Moreover, we find useful the reference n°144 to better explain how cathecolamine produced during physical activity may affect immune response. 

We appreciated your suggested review clarifying BDNF role and we discussed it (ref n°137) . However, how physical activity enhance BDNF level are still unclear. We took the opportunity to revise current existent hypotheses underlying this mechanism in the appropriate section of the manuscript (ref n°141-142).

MD diet probably is the more appropriate diet regimen in providing the necessary calory intake as well as in modulating inflammatory response with higher patient’s compliance. Therefore, we appreciated your advice to explain how MD might affect MS course, indirectly ameliorating chronic comorbidities or directly the inflammatory response (ref n°38). 

We are sorry for  mistakes at 108, 258 and 330. These are now fixed.

According to your precious advice, we also produced a smart figure (Figure 1 at the bottom of manuscript) resuming the changes in inflammation pathways of the immune and innate response drived by diets and physical activity. 

We updated the bibliography. 

Finally, the manuscript underwent to English-editing by English speaking professor.

Reviewer 2 Report

Timely and well written narrative review which summarizes the role of nutritional lifestyle and physical activity in people with multiple sclerosis. I do not have any minor or major issues.

Author Response

We are grateful for your report.